# Circulating Alpha-Tocopherol Levels, Bone Mineral Density, and Fracture: Mendelian Randomization Study

**DOI:** 10.3390/nu13061940

**Published:** 2021-06-05

**Authors:** Karl Michaëlsson, Susanna C. Larsson

**Affiliations:** 1Unit of Medical Epidemiology, Department of Surgical Sciences, Uppsala University, SE-751 85 Uppsala, Sweden; susanna.larsson@ki.se; 2Unit of Cardiovascular and Nutritional Epidemiology, Institute of Environmental Medicine, Karolinska Institutet, SE-171 77 Stockholm, Sweden

**Keywords:** alpha-tocopherol, bone mineral density, fracture, Mendelian randomization, vitamin E

## Abstract

Recent cohort studies indicate a potential role of the antioxidant α-tocopherol in reducing bone loss and risk of fractures, especially hip fractures. We performed a Mendelian randomization investigation of the associations of circulating α-tocopherol with estimated bone mineral density (eBMD) using heel ultrasound and fractures, identified from hospital records or by self-reports and excluding minor fractures. Circulating α-tocopherol was instrumented by three genetic variants associated with α-tocopherol levels at *p* < 5 × 10^−8^ in a genome-wide association meta-analysis of 7781 participants of European ancestry. Summary-level data for the genetic associations with eBMD in 426,824 individuals and with fracture (53,184 cases and 373,611 non-cases) were acquired from the UK Biobank. Two of the three genetic variants were strongly associated with eBMD. In inverse-variance weighted analysis, a genetically predicted one-standard-deviation increase of circulating α-tocopherol was associated with 0.07 (95% confidence interval, 0.05 to 0.09) g/cm^2^ increase in BMD, which corresponds to a >10% higher BMD. Genetically predicted circulating α-tocopherol was not associated with odds of any fracture (odds ratio 0.97, 95% confidence interval, 0.91 to 1.05). In conclusion, our results strongly strengthen a causal link between increased circulating α-tocopherol and greater BMD. Both an intervention study in those with a low dietary intake of α-tocopherol is warranted and a Mendelian randomization study with fragility fractures as an outcome.

## 1. Introduction

Worldwide, 1.6 million hip fractures are estimated to occur each year [1] at an average of 80 years of age. Hip fractures and other types of fragility fractures have, besides heavy costs for the society, a profound impact on quality of life; only one-third of these fracture patients regain their pre-fracture level of function [1]. There is also a higher risk of death after the fracture event [2], and especially vulnerable are older men [2]. Scandinavia has one of the highest incidences of fragility fractures in the world, with a lifetime cumulative incidence of 50% in women and 25% in men [3,4]. The risk of hip fracture, the most devastating fragility fracture, increases 44-fold in Swedish women from 55 to 85 years of age, so that the lifetime risk of hip fracture is 25% in women and 12% in men [4]. Bone mineral density (BMD) is a strong determinant of future risk of hip fracture. Hip fracture rates are more than doubled for each standard-deviation lower BMD at the hip [5], whereas the association between other BMD and fractures sites is generally less strong [5,6]. 

Fracture risk and bone loss are determined both by genotype and by environmental factors, where the importance of lifestyle factors increases with advancing age [7,8]. Of importance, there are now strong indications that oxidative stress is a central biological mechanism for bone cell senescence, skeletal aging, and loss of BMD [9,10,11], an important determinant contributing to fracture risk [5,12]. 

By decreasing the generation of free radicals leading to lower oxidative stress, the risk of low BMD in the elderly might be reduced [9,13,14]. The antioxidant α-tocopherol is the most potent form of vitamin E that has the potential to scavenge free radicals and has been proposed to favorably affect BMD [9,13,14,15,16]. However, some experimental evidence indicates that excessive intakes of α-tocopherol decrease bone mass in mice [17], a finding not confirmed by more recent evidence [18,19]. Studies in humans on α-tocopherol in relation to bone health are limited and include only observational study designs. A majority of [14,19,20,21,22] but not all [23,24] previous observational studies indicate that low dietary intakes and low serum levels of α-tocopherol are associated with lower BMD and an increased risk of fracture, especially hip fractures. Of importance, in many European countries, the mean α-tocopherol intake is below the recommended levels [25].

Genetic variants that specifically affect a biomarker can be used as instrumental variables (proxies) for the biomarker to determine whether the biomarker is causally associated with the outcome. This approach, recognized as Mendelian randomization, has previously been applied to assess the associations of lifelong circulating metabolite concentrations with various diseases [26,27,28] but has not been used to examine serum concentrations of α-tocopherol and bone phenotypes.

Thus, the Mendelian randomization (MR) design can overcome residual confounding and other biases in observational studies, thereby strengthening causal inference for an exposure–outcome association by leveraging genetic variants as proxy indicators for an exposure [29]. In this study, we used the MR design to examine the associations of genetically predicted circulating α-tocopherol levels with BMD and risk of any type of fracture. 

## 2. Methods

### 2.1. Selection of Genetic Variants

We used three uncorrelated single-nucleotide polymorphisms (SNPs) related to circulating α-tocopherol concentrations at the level of genome-wide significance (*p* < 5 × 10^−8^) in a meta-analysis of three genome-wide association studies comprising 7781 individuals of European ancestry [30]. The association estimates were adjusted for age, cancer status, and body mass index and, because it is well recognized that vitamin E levels are influenced by blood lipids, additional adjustment was made for total and high-density lipoprotein cholesterol. The three SNPs included rs964184 on 11q23.3 close to *BUD13*, *ZNF259*, and *APOA1/C3/A4/A5* (*p* = 7.8 × 10^−12^), rs2108622 on 19pter-p13.11 close to *CYP4F2* (*p* = 1.4 × 10^−10^), and rs11057830 on 12q24.31 close to *SCARB1* (*p* = 8.2 × 10^−9^). These genetic variants explained around 1.7% of the variation in log serum α-tocopherol levels [30]. Mean (±standard deviation) α-tocopherol levels ranged from 11.9 (±3.4) mg/L to 19.1 (±9.7) mg/L in the included GWASs [30]. 

### 2.2. Summary-Level Data for Outcomes

Summary-level data for the associations between the α-tocopherol-related SNPs and estimated BMD (eBMD), using heel quantitative ultrasound in 426,824 participants and fracture (53,184 cases and 373,611 non-cases) were taken from GWASs based on data from UK Biobank [31]. The heel quantitative ultrasound method can measure BMD to a similar degree as dual-energy X-ray absorptiometry and is an inexpensive, easy to implement, and radiation-free technique [32]. Mean (±standard deviation [SD]) eBMD levels were 0.56 ± 0.12 g/cm^2^ in men and 0.51 ± 0.11 g/cm^2^ in women. Fractures were identified by hospital Episodes Statistics using ICD-10 codes (*n* = 20,122) and questionnaire-based self-reported data (*n* = 48,818). Omitted were fractures of the face and skull, hands and feet, atypical femoral fractures, periprosthetic fractures, restored fractures, and pathological fractures caused by malignancy [31]. The genetic estimates were adjusted for ancestry-informative genetic principal components 1 to 20, genotyping array, assessment center, sex, and age.

### 2.3. Two-Sample Summary-Level MR Analysis

A ratio estimate for each of the three SNPs was computed by dividing the beta coefficient for the SNP–eBMD or the SNP–fracture association by the beta coefficient for the SNP–α-tocopherol association. These ratio estimates were pooled in a fixed-effects inverse-variance weighted model to obtain the MR estimates per one SD increment of the association of serum α-tocopherol with BMD and fracture risk. The SD was estimated from a population-based Swedish cohort of men (*n* = 2047; https://www.pubcare.uu.se/ulsam/, (accessed on 1 June 2021)). Mean (±SD) serum α-tocopherol levels were 13.1 (±3.5) mg/L and 2.5 (±0.25) mg/L on the normal and log-transformed scale, respectively. Stata software (version 14.0) was used for the analyses. 

### 2.4. Pleiotropy Assessment

We searched the PhenoScanner database [33] to assess whether the α-tocopherol-associated SNPs were associated with known risk factors for low BMD or fracture risk. We considered the following factors: body mass index, fat-free soft tissue body mass, height, type 2 diabetes, smoking, alcohol consumption, and steroid hormones, including estrogens, testosterone, and cortisol. 

## 3. Results

The characteristics of the three SNPs associated with α-tocopherol and their associations with eBMD and fracture risk are shown in Appendix A. Two of the three α-tocopherol-associated SNPs were strongly associated with eBMD (Figure 1). Genetically predicted one-SD increment in circulating α-tocopherol levels was associated with 0.07 (95% confidence interval, 0.05 to 0.09) g/cm^2^ higher eBMD (*p* < 0.001), which corresponds to >10% higher eBMD. 

Genetically predicted circulating α-tocopherol was not associated with odds of fracture (odds ratio 0.97, 95% confidence interval, 0.91 to 1.05; *p* = 0.48) (Figure 2). None of the three SNPs was associated with known risk factors for low BMD and fracture at *p* <0.01. 

## 4. Discussion

This MR investigation showed that genetically proxied higher circulating α-tocopherol levels were clearly and independently associated with greater eBMD. Our results are consistent with several observational studies that found a positive correlation between serum α-tocopherol and higher BMD [14,19,20,21,22]. We were unable to find a significant association between genetically predicted α-tocopherol levels and fracture risk. In cohort studies, strongest associations have been found with hip fracture as an outcome, not yet available for MR study designs. 

Previous cohort studies reveal that both higher blood levels of α-tocopherol and higher dietary α-tocopherol intake are related to higher BMD and reduced risk of osteoporosis and fractures, including hip fractures [14,19,20,21]. At the time of the fracture event, hip fracture patients have been shown to have lower serum α-tocopherol concentrations compared with controls [22], and serum α-tocopherol concentrations after the hip fracture event are associated with lower levels of inflammatory markers [14,34]. In addition, higher circulating α-tocopherol concentrations are related to improved physical performance after a hip fracture event [14,35]. Interestingly, no association was found between bone mineral density and serum vitamin E concentrations in the Women’s Health Initiative Study [14,24]. Of notice, however [14], the women in that study had a mean total intake of vitamin E (including supplements) of about 30 mg/day [24], which is three times higher than the recommended intake. 

Studies in animals have shown that supplementation with α-tocopherol improves fracture healing and may also improve osseointegration of metallic implants [14,15,36,37,38,39] These findings contrast with results in a Japanese study [17], which reported that high α-tocopherol intake was harmful for bone by stimulating bone resorption followed by a decrease in bone mass. The investigators supplemented young rodents (mice and rats) with vitamin E equivalent to a 30-fold higher dose than the normal intake recommended for these species [14,40]. Continued high-dose administration of α-tocopherol might be toxic [41] and results in appetite loss, in turn leading to impaired weight and skeletal growth. A high vitamin E intake may also adversely influence the use of vitamin D, leading to decreased bone mass [42]. In contrast, other experimental evidence indicates that a high vitamin E dose had positive effects on bone health in rodents [15,18,19], and thus, the findings presented in the study by Fujita and colleagues [17] displaying an adverse effect on bone health by high dosing of vitamin E could not later be replicated. Indeed, anti-osteoporotic properties of vitamin E have been demonstrated in different animal models [43]. Vitamin E modulates the levels of inflammatory mediators and reactive oxygen species, acting systemically and locally, with a potent regulatory role in bone metabolism [43]. Specifically, vitamin E plays an essential role in oxidative stress signaling, with effects on the receptor activator of nuclear factor kappa-B (RANK)/receptor activator of nuclear factor kappa-B ligand (RANKL)/osteoprotegerin (OPG) and Wnt/β-catenin systems, affecting osteoclast and osteoblast activity [43]. There are strong indications of an effect of oxidative stress on bone senescence [9,10,11,44,45,46]. Supporting the view that vitamin E dose is of importance also in humans, a meta-analysis of placebo-controlled randomized trials revealed that low-dosage vitamin E supplements can reduce all-cause mortality [47], whereas high-dosage vitamin E supplements, with a mean dose of 400 mg/day corresponding to 40 times the recommended intake [47], may in contrast lead to higher death rates [14,47]. 

A strength of our study is the MR design that reduced potential confounding and reverse causation bias and thus strengthened the causal inference in the association between circulating α-tocopherol and BMD. The current study was confined to participants of European origin. Thus, it is not likely that our findings were affected by a population stratification bias. The genetic instrument was not developed in the UK Biobank, while this large cohort was used for instrument–outcome association analyses to estimate a causal effect of blood levels of α-tocopherol on the outcomes BMD and fractures. An overlap in participants between the instrument-development and outcome samples can cause bias towards the risk factor–outcome association [48].

A limitation of our study is that few SNPs were available as genetic instruments for α-tocopherol, which limited the possibility to assess possible pleiotropy (i.e., where one genetic variant influences multiple phenotypes) through robust MR methods such as MR-Egger regression. Although none of the SNPs was associated with known risk factors for low BMD and fracture risk, the variants in the *BUD13/ZNF259/APOA5* and *CYP4F2* gene regions were associated with circulating phylloquinone (at *p* = 6 × 10^−8^ and *p* = 8.8 × 10^−7^, respectively) [49]. Nevertheless, there is little evidence that phylloquinone affects BMD [50,51,52]. A major limitation for the fracture analysis is the hitherto relatively young age of fracture cases in the UK Biobank, inclusion of self-reported fractures [53], and a mixture of different types of fractures. An MR study focusing on fragility fractures, especially hip fractures, would be of interest, but such GWAS data are not available. 

## 5. Conclusions

In conclusion, our results strengthen the view that increasing circulating α-tocopherol is associated with higher BMD. A randomized clinical trial to investigate the effect on BMD and fracture risk of moderate doses corresponding to the daily recommended intake of vitamin E is warranted.

## Figures and Tables

**Figure 1 nutrients-13-01940-f001:**
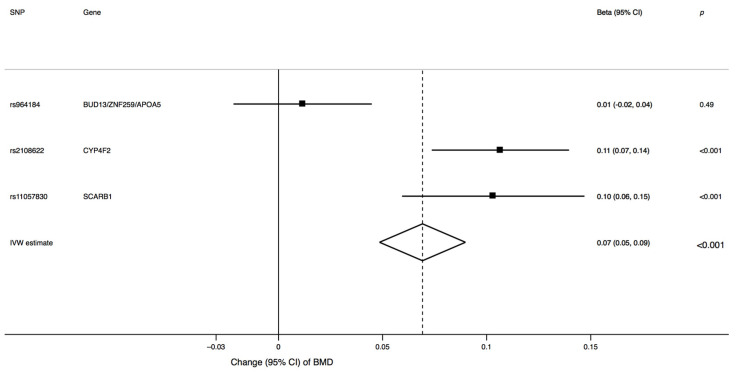
Association between genetically predicted one-standard-deviation increase in circulating α-tocopherol and BMD. CI, confidence interval; BMD, bone mineral density; IVW, inverse, variance weighted; SNP, single-nucleotide polymorphism.

**Figure 2 nutrients-13-01940-f002:**
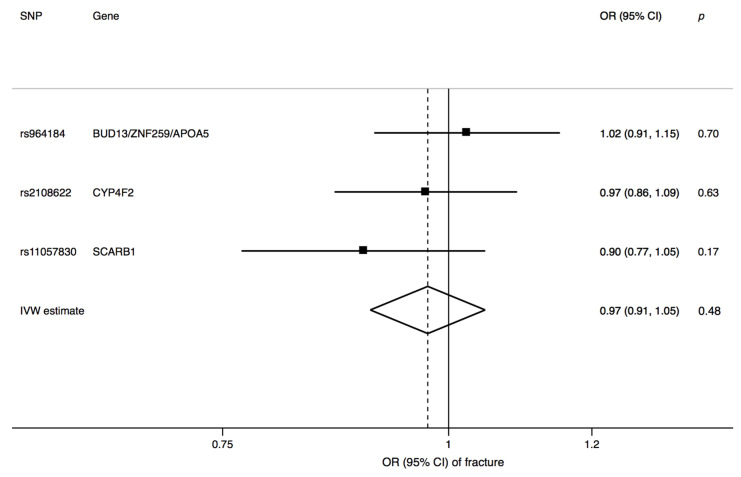
Association between genetically predicted one-standard-deviation increase in circulating α-tocopherol and fracture risk. CI, confidence interval; IVW, inverse, variance weighted; OR, odds ratio; SNP, single-nucleotide polymorphism.

## Data Availability

All data used in this study are publicly available summary statistics data, with relevant data available from cited studies. The summary statistics data analyzed in this study are available on request from the corresponding author.

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
