# Peer review of "Circulating Alpha-Tocopherol Levels, Bone Mineral Density, and Fracture: Mendelian Randomization Study"

_nutrients, 2021, doi:10.3390/nu13061940_

Round 1

Reviewer 1 Report

I would  like to  see  the  F-stat  and power calculated  for  all  of  the  SNPs, in  simple  way  to be digestible  for  the  non-expert (Supp  file). SD/R2 for the IVs has to be mentioned. All  the  details  re  the  beta, se, effect allele,  other  allele  and  EAF need  to  be  mentioned in  the  Supp file for the IVs.  Please clearly elaborate how you have ruled out the chance of pleiotropy at PhenoScanner database. I would like to see further sensitivity analysis such as WM and egger

Author Response

Reviewer 1

Point 1: I would like to see the F-stat and power calculated for all of the SNPs in simply way to be digestible for the non-expert (Supp file). SD/R2 for the IVs has to be mentioned.

Response 1: We agree that the F-statistic for each SNP may have been of value for other researchers conducting MR studies. Unfortunately, information about the SD for alpha-tocopherol in the GWAS on alpha-tocopherol is not available so we are unable to calculate the R2 value, F-statistic, and power for each SNP.

Point 2: Details regarding the beta, se, effect allele, other allele and EAF need to be mentioned in the Supp file for the IVs.  Please clearly elaborate how you have ruled out the chance of pleiotropy at PhenoScanner database. I would like to see further sensitivity analysis such as WM and egger.

Response 2: As recommended, we have now proved the beta coefficient, SE, effect and other allele for each SNP used (Supplemental table 1). We agree that sensitivity analyses such as the weighted median, MR-Egger, and weighted mode methods can be useful complementary methods in MR analyses, at least when there are multiple SNPs (more than 3-10 SNPs) available for the exposure. In this MR study on alpha-tocopherol, we only had 3 genetic instruments (SNPs) and therefore those alternative methods were not applicable.

Reviewer 2 Report

The authors used the MR design to examine the associations of genetically predicted circulating α-tocopherol levels with BMD and risk of any type of fracture. The paper is nicely written but lacks detail, especially for those less familiar with MR studies.

Can the authors please explain why they adjusted the association estimates for age, cancer status, BMI and both total & HDL cholesterol. Should you also adjust for calcium and vitamin D?

What were the genetic principal components that the genetic estimates were adjusted for, in addition to age & sex?

Please provide more detail on your pleiotropy section. Did you do anything with the factors that you considered? Should you also have presented alternative approaches such as MR-Egger, weighted median or weighted mode estimates? You note this in your limitations - do you feel your results may be biased towards a protective effect?

In the 3rd paragraph of the Discussion, you state' vitamin A equivalent' - do you mean vitamin E - I couldn't access ref 32.

Author Response

Reviewer 2

Point 1: The authors used the MR design to examine the associations of genetically predicted circulating α-tocopherol levels with BMD and risk of any type of fracture. The paper is nicely written but lacks detail, especially for those less familiar with MR studies.

Response 1: Thank you for your comments. We have now extended the introduction part of the manuscript for a better guidance to the problem of fragility fractures and why an MR design can improve our understanding of causality.

Point 2: Can the authors please explain why they adjusted the association estimates for age, cancer status, BMI and both total & HDL cholesterol. Should you also adjust for calcium and vitamin D?

Response 2: The adjustments were made in the original GWAS study, and not by us. As we are only using summary-level data (that is, beta coefficients and SEs) taken from the original GWAS studies, we are unable to change the adjustment factors, such as including adjustment for vitamin D and calcium. However, adjustment for vitamin D and calcium would not change our results and is unnecessary as none of the genetic variants that affect alpha-tocopherol levels are related to the levels of vitamin D and calcium, and previous MR studies have shown that neither vitamin D nor calcium within the normal range are causally associated with BMD and fracture risk.

Point 3: What were the genetic principal components that the genetic estimates were adjusted for, in addition to age & sex?

Response 3: Adjustment was made for 1 to 20 ancestry informative principal components. We have added this information in the text.

Point 4: Please provide more detail on your pleiotropy section. Did you do anything with the factors that you considered? Should you also have presented alternative approaches such as MR-Egger, weighted median or weighted mode estimates? You note this in your limitations - do you feel your results may be biased towards a protective effect?

Response 4: We agree that sensitivity analyses such as the weighted median, MR-Egger, and weighted mode methods can be useful complementary methods in MR analyses, at least when there are multiple SNPs (more than 3-10 SNPs) available for the exposure. In this MR study on alpha-tocopherol, we only had 3 genetic instruments (SNPs) and therefore those alternative methods were not applicable.

Point 5: In the 3rd paragraph of the Discussion, you state' vitamin A equivalent' - do you mean vitamin E - I couldn't access ref 32.

Response 5: This was an error. The test should read “vitamin E equivalent …”

Reviewer 3 Report

The article is written in clear, specific and concise language, but authors should check punctuation, for example: page 1 “ …bone mineral density (BMD),[3-5] an important …” and check page 4 in line "The investigators  supplemented young rodents (mice and rats) with vitamin A equivalent ... " should there be vitamin A or E?

Author Response

Reviewer 3

Point 1: The article is written in clear, specific and concise language, but authors should check punctuation, for example: page 1 “ …bone mineral density (BMD),[3-5] an important …” and check page 4 in line "The investigators  supplemented young rodents (mice and rats) with vitamin A equivalent ... " should there be vitamin A or E?

Response 1: Thank you for the suggested edits and you are right, we should have written vitamin E. Changes at the suggested places have now been made.

Reviewer 4 Report

Line 6 of the abstract, I believe you only need one hyphen between alpha and tocopherol

Author Response

Reviewer 4

Point 1: Line 6 of the abstract, I believe you only need one hyphen between alpha and tocopherol.

Response 1: We have now deleted the extra hyphen. Thank you for noticing this error.